# Use of primary care and other healthcare services between age 85 and 90 years: longitudinal analysis of a single-year birth cohort, the Newcastle 85+ study

Mohammad Esmaeil Yadegarfar,[1,2] Carol Jagger,[1,2] Rachel Duncan,[1,2] Tony Fouweather,[2] Barbara Hanratty,[1,2] Stuart Parker,[1,2] Louise Robinson[1,2]

[1]Campus for Ageing and Vitality, Institute of Ageing, Newcastle University, Newcastle upon Tyne, UK
[2]Campus for Ageing and Vitality, Institute of Health and Society, Newcastle University, Newcastle upon Tyne, UK

**Correspondence to**
Dr Louise Robinson;
a.l.robinson@ncl.ac.uk

## ABSTRACT

**Objective**  To describe, using data from the Newcastle 85+ cohort study, the use of primary care and other healthcare services by 85-year-olds as they age.

**Design**  Longitudinal population-based cohort study.

**Setting**  Newcastle on Tyne and North Tyneside, UK.

**Participants**  Community dwelling and institutionalised men and women recruited through general practices (n=845, 319 men and 526 women).

**Results**  Contact was established with 97% (n=1409/1459) of eligible 85-year-olds, consent obtained from 74% (n=1042/1409) and 851 agreed to undergo the multidimensional health assessment and a general practice medical records review. A total of 845 participants had complete data at baseline for this study (319 male, 526 female), with 344 (118 male, 226 female) reinterviewed at 60 months. After adjusting for confounders, all consultations significantly increased over the 5 years (incidence rate ratio, IRR=1.03, 95% CI 1.01 to 1.05, P=0.001) as did general practitioner (GP) consultations (IRR=1.03, 95% CI 1.01 to 1.05, P=0.006). Significant increases were also observed in inpatient and day hospital use over time, though these disappeared after adjustment for confounders.

**Conclusions**  Our study of primary, secondary and community care use by the very old reveals that, between the ages of 85 and 90 years, older people are much more likely to consult their GP than any other primary healthcare team members. With a rapidly ageing society, it is essential that both current and future GPs are appropriately skilled, and adequately supported by specialist colleagues, as the main healthcare provider for a population with complex and challenging needs.

## Strengths and limitations of this study

► This study provided unique opportunity to analyse a large cohort of older adults' use of healthcare services extracted from general practitioner medical records avoiding potential bias and inaccuracy emanating from self-reported or extracted research databases.

► Information on healthcare professional and consultation type provided much needed insight about the needs of this age group in both primary and secondary care settings.

► The absence of any information on consultation length and complexity precludes comment on the detailed nature of the increased workload in primary care.

► Our estimates of healthcare use are conservative, as consultations were analysed for 12 months prior to each interview and not the 12 months leading to death when healthcare use can be intensive.

of people aged 85 years and over[2] revealed multimorbidity to be the norm,[3] yet the majority remain able to live independently although with family support.[3 4] Alongside multimorbidity, increasing age carries a greater risk of physical frailty[5 6] and cognitive impairment and dementia.[7] Between 25% and 50% of those over 85 years are estimated to be frail,[8] placing them at increased risk of death and disability and admission to hospital and long-term care.[9] Dementia contributes a bigger disease burden than other long-term illness such as cancer or stroke, with considerable care costs, especially in the last year of life.[7 10]

Primary care services are central to the provision of healthcare in many developed countries, including the UK. Family physicians, or general practitioners (GPs), and their teams provide the first point of contact for patients, diagnose disease, monitor long-term conditions and have a pivotal role in

## INTRODUCTION

Our society is rapidly ageing. The fastest growing sector of our population is *the very old*, those aged 85 years and over; between 2015 and 2035, the older population of England and Wales (aged 65 years and over) is projected to increase by 48%, whereas numbers aged 85 years and older will rise by 113%.[1] Findings from the first UK study to successfully recruit and retain a large cohort

disease prevention. It has long been acknowledged that primary care-led healthcare systems deliver more efficient and equitable services,[11] with healthier, more satisfied patients, for lower cost and with fewer inequalities in both health and access to care.[12 13] With a rapidly ageing population, the resulting larger proportion experiencing multimorbidity, cognitive decline and frailty, could place considerable pressures on health and social care provision, especially primary and community care services, in a system where the former is the first and main source of healthcare. However, in the UK, primary care services are already almost at 'saturation point' with substantial increases in consultation rates and consultation duration with the population as a whole.[14]

The aim of this paper is to describe, using data from the Newcastle 85+ study, the use of primary and secondary care services by a cohort of the very old as they age over a 5-year period.

## METHODS

The Newcastle 85+ study is a prospective observational longitudinal study of a 1921 birth cohort who turned 85 during 2006.[2 3] Potential participants were recruited from general practitioner (GP) registered patient lists in Newcastle on Tyne and North Tyneside: contact was established with 97% (n=1409/1459) of eligible 85-year-olds. Consent was obtained from 74% (n=1042/1409); 851 agreed to undergo detailed multidimensional health assessment (MDHA) and a general practice medical records review (GPRR); three consented to MDHA only; 188 consented to GPRR only and 358 declined all participation. Analysis of response, attrition and comparison with the national birth cohort have already been published.[2 3]

As part of their GPRR, participant's primary healthcare use was recorded for the 12 months prior to their assessment interview (baseline, 36 and 60 months). At baseline and 36 months, information gathered included consultations with 16 different professionals seen during these periods. Data for each participant were summarised in three ways: total number of consultations with each of the professionals separately; total number of consultations with any primary care professional (GP, GP out of hours, practice nurse/practitioner/healthcare assistant (HCA), community nurse, health visitor) and total number of visits to any of the 16 professionals (table 1). At 60 months, only GP and non-GP primary care consultation were identified with remaining professionals (GP out of hours, practice nurse/practitioner/HCA, community nurse, health visitor) as at baseline and 36 months (table 1).

Additional information on secondary care use was collected for all participants at interview: inpatient, day hospital (total number of days spent in the 12 months prior to interview); outpatient, and accidents and emergency (A&E) (total number of visits in the 3 months prior to interview) (table 1). Sociodemographic and health

characteristics of participants were collected at baseline, 36 and 60 months follow-up.

### Statistical analysis

Baseline sociodemographic (living status, self-rated health, education) and health characteristics (Mini-Mental State Examination (MMSE), Geriatric Depression score (GDS), disability, disease group count) of participants and sex differences were analysed using $\chi^2$ test for categorical data and Mann-Whitney U test for count data. Trends in healthcare use over time were analysed by negative binomial regression as the data were overdispersed (variance much greater than mean). Zero-inflated negative binomial regression models were used for outcomes where there were high numbers of zero consultations. Final models were adjusted for sex, sociodemographic and health characteristics. Confounding factors (living status, self-rated health, MMSE, GDS, disability and disease count) were measured at multiple time points (apart from education) and values were updated in models. Time trends were reported as incidence rate ratios (IRR). Primary and secondary care usage were analysed in the overall sample and in participants who took part at all three time points (baseline, 36, 60 months). All analyses were undertaken in Stata V.12.0 (Stata).

## RESULTS

At baseline, the study comprised 845 participants (319 men and 526 women), of whom 10.2% (n=86) were living in residential care, 12.5% (n=105) had moderate or severe cognitive impairment (MMSE score 18 or less), 6.3% (n=53) had severe disability and 18.6% (n=157) had four or more diseases (table 2).

Between ages 85 and 90 years, the mean number of all consultations increased significantly by 2.9 extra consultations (P<0.001) and the mean number of GP consultations by 1.6 (P<0.001) (table 3). There was an increase in primary care consultations of 0.8 consultations between ages 85 and 88 of which the majority (0.6 consultations) were with the GP (table 3). The same pattern of consultation use over time was found when the analysis was confined to participants who were alive at all three time points (table 3). After adjustment for confounding factors, there was a significant increase over the 5 years in all consultations (IRR=1.03, 95% CI 1.01 to 1.05, P=0.001) and GP consultations (IRR=1.03, 95% CI 1.01 to 1.05, P=0.006) (figure 1).

Analysis of the change in secondary care use between ages 85 and 90 years revealed a non-significant increase in mean inpatient days of 3.8 days (P=0.071), although when restricted to participants who survived to age 90, the mean inpatient days increased by 5 days (P=0.010) (table 3). No significant changes in mean number of days as a day patient, outpatient or visits to A&E were found (table 3). After adjustment for confounding factors, no significant trends over time were found for any of the secondary healthcare use (inpatient days, day hospital,

**Table 1** Description of outcomes and confounding factors included

| Variable | Variable description | Variable type |
|---|---|---|
| **Primary healthcare use** | | |
| GP practice | | |
| GP practice out of hours | | |
| Practice nurse/ practitioner/HCA | | |
| Community nurse | | |
| Health visitor | | |
| Dietician | | |
| Phlebotomist | | |
| Other | This variable records all consultations participants had with a healthcare professional 12 months prior to each MDHA at each time point. | Outcome |
| Not specified | | |
| Clerical | | |
| Pharmacist/pharmacy technician | | |
| Chiropodist/podiatrist | | |
| Physiotherapist | | |
| Counsellor/practice counsellor | | |
| Psychiatrist | | |
| Mental health worker | | |
| **Secondary healthcare use** | | |
| Inpatient | Time spent by participants for each different type of hospital admission. | |
| Day patient | Days spent during the 12 month prior to MDHA. | |
| Outpatient | | Outcome |
| A&E | Number of visits during the 3 months prior to MDHA. | |
| **Time** | This is a continuous measure of time in years from the start of baseline interview to participant's death. | Covariate |
| **Living status** | | |
| Alone in community | | |
| Not alone in community | Participant's living arrangements at each MDHA. | Covariate |
| Institutional living | | |
| **Self-rated health** | | |
| Excellent/very good | | |
| Good | Participant's perception of their general health on a five-point scale recoded into three categories. | Covariate |
| Fair/poor | | |
| **MMSE** | | |
| Normal (26–30) | | |
| Mild (22–25) | | |
| Mod (18–21) | Participant's categorised MMSE scores at each MDHA. | Covariate |
| Severe (0–17) | | |

Continued

**Table 1** Continued

| Variable | Variable description | Variable type |
|---|---|---|
| **GDS** | | |
| No depression | | |
| Mild | Categorised GDS collected at each MDHA. | Covariate |
| Severe | | |
| MMSE<15 | | |
| **Categorised disability** | | |
| None | | |
| 1–6 | Categorised disability score based on activities of daily living, collected at each MDHA. | Covariate |
| 7–12 | | |
| 13–17 | | |
| **Disease groups** | | |
| 0 | | |
| 1 | Categorised disease groups (maximum eight). Eight disease groups were identified with each scored 1 if the participants had a GP diagnosis of said disease at each GPRR. Disease groups included the following: arthritis, cancer, cardiac disease, cerebrovascular disease, diabetes mellitus, hypertension, respiratory disease and cognitive Impairment. | Covariate |
| 2–3 | | |
| 4+ | | |

A&E, accidents and emergency; GDS, Geriatric Depression Score; GP, general practitioner; GPRR, GP record review; HCA, healthcare assistant; MDHA, multidimensional health assessment; MMSE, Mini-Mental State Examination.

outpatient visits, A&E visits) (figure 2). Conclusions remained unchanged when analysis was confined to participants who survived the 5 years (data not shown).

## DISCUSSION

Our study suggests that over the age of 85 years, older people are increasingly likely to consult their GP within the primary care team for their healthcare needs; indeed, by the age of 90 years, most primary care consultations are with the GP. In contrast, no significant changes were found in the use of secondary care services, including A&E and outpatient clinics. These patterns remained after adjustment for changing sociodemographic factors (including admission to care homes and health factors such as multimorbidity and declining cognitive function). These findings help to explain the increasing workload in UK primary care; if GPs are consulting with the growing and increasingly complex population of 85-year-olds, who show no increase in use of secondary care services.[14]

### Strengths and limitations

This study analysed a unique dataset on a large cohort of older adults' use of services. The extraction of data directly from GP medical records is a key strength, as it

**Table 2** Baseline sociodemographic and health characteristics of the 85+ study participants

| Characteristic * | Males (319) %(n) | Females (526) | All (845) | P value |
|---|---|---|---|---|
| Living status | | | | |
| Alone in community | 39.5 (126) | 64.0 (336) | 54.7 (462) | <0.001 |
| Not alone in community | 54.2 (173) | 23.4 (123) | 35.1 (296) | |
| Institutional living | 6.3 (20) | 12.6 (66) | 10.2 (86) | |
| Self-rated health | | | | |
| Excellent/very good | 43.9 (137) | 37.7 (193) | 40.1 (330) | 0.152 |
| Good | 36.5 (114) | 38.3 (196) | 37.6 (310) | |
| Fair/poor | 19.6 (61) | 24.0 (123) | 22.3 (184) | |
| Education | | | | |
| 0–9 years | 62.3 (195) | 65.7 (339) | 64.4 (534) | 0.576 |
| 10–11 years | 24.6 (77) | 21.7 (112) | 22.8 (189) | |
| 12+ years | 13.1 (41) | 12.6 (65) | 12.8 (106) | |
| MMSE | | | | |
| Normal (26–30) | 71.9 (228) | 71.1 (371) | 71.4 (599) | 0.113 |
| Mild (22–25) | 18.3 (58) | 14.8 (77) | 16.1 (135) | |
| Mod (18–21) | 3.5 (11) | 6.9 (36) | 5.6 (47) | |
| Severe (0–17) | 6.3 (20) | 7.3 (38) | 6.9 (58) | |
| GDS | | | | |
| No depression | 79.7 (247) | 71.4 (360) | 74.6 (607) | 0.066 |
| Mild | 9.0 (28) | 13.9 (70) | 12.0 (98) | |
| Severe | 6.8 (21) | 8.5 (43) | 7.9 (64) | |
| MMSE<15 | 4.5 (14) | 6.2 (31) | 5.5 (45) | |
| Categorised disability | | | | |
| None | 31.6 (100) | 16.3 (85) | 22.1 (185) | <0.001 |
| 1–6 | 52.4 (166) | 57.5 (300) | 55.5 (466) | |
| 7–12 | 11.7 (37) | 18.8 (98) | 16.1 (135) | |
| 13–17 | 4.4 (14) | 7.5 (39) | 6.3 (53) | |
| Disease groups† | | | | |
| 0 | 6.6 (21) | 4.2 (22) | 5.1 (43) | 0.448 |
| 1 | 19.4 (62) | 21.5 (113) | 20.7 (175) | |
| 2–3 | 55.5 (177) | 55.7 (293) | 55.6 (470) | |
| 4+ | 18.5 (59) | 18.6 (98) | 18.6 (157) | |

*Data available at each time point for all characteristics except education.
†For diseases included, see table 1.
GDS Geriatric Depression Score; MMSE, Mini-Mental State Examination.

avoids the potential bias and inaccuracies of data that are self-reported or extracted from research databases. The absence of any information on consultation length and complexity precludes comment on the detailed nature of the increased workload in primary care. One limitation of our data is the less fine-grained coding of professionals consulted at 60 months to reduce data collection time. This meant that increases in consultations by individual primary care professionals could not be compared over the whole 5-year period between ages 85 and 90 years. However, since the vast majority of the increases in consultations between ages 85 and 88 years were with the GP, it seems unlikely that this trend would be reversed in favour of other professionals. Consultations were analysed for the 12 months prior to each interview, therefore excluding data on those who had not been interviewed at that time, mostly due to death. Our estimates of healthcare use are therefore conservative since healthcare use at end of life can be intensive in the 12 months leading to death.

Such findings are of considerable concern for the UK in terms of ensuring that both the current and future

**Table 3** Mean number of consultations (healthcare use) at each time point of the study for all participants, by sex

| All participants (n=845) | Baseline (n=845) | 36 months (n=485) | 60 months (n=344) | P value |
|---|---|---|---|---|
| | Mean (SD) | | | |
| All consultations | 10.4 (7.7) | 11.4 (8.3) | 13.3 (13.6) | <0.001 |
| Primary care consultations | 9.8 (7.5) | 10.6 (7.8) | –* | 0.064 |
| GP | 5.9 (4.8) | 6.5 (5.9) | 7.5 (6.5) | <0.001 |
| GP out of hours service | 0.1 (0.5) | 0.2 (0.8) | –* | 0.575 |
| Practice nurse/practitioner/HCA† | 2.8 (3.0) | 2.6 (3.0) | –* | 0.634 |
| Community nurse† | 1.0 (3.9) | 1.1 (3.0) | –* | 0.823 |
| Clerical | 0.3 (0.7) | 0.3 (1.6) | 5.8 (10.7) | <0.001 |
| Pharmacist/pharmacy technician | 0.1 (0.3) | 0.0 (0.0) | 0.0 (0.3) | 0.693 |
| All other consults | 0.2 (0.7) | 0.5 (1.1) | 0.0 (0.5) | <0.001 |
| Inpatient | 3.6 (15.3) | 4.6 (14.0) | 7.4 (18.6) | 0.071 |
| Day patient | 0.2 (0.9) | 0.2 (0.6) | 0.1 (0.4) | 0.027 |
| Outpatient | 0.6 (1.8) | 0.6 (1.2) | 0.6 (1.9) | 0.974 |
| A&E‡ | 0.1 (0.3) | 0.1 (0.4) | 0.1 (0.4) | 0.500 |
| **Participants alive at 60 months (n=344)** | **(n=344)** | **(n=344)** | **(n=344)** | |
| All consultations | 9.9 (6.6) | 10.8 (8.1) | 13.3 (13.6) | <0.001 |
| Primary care consultations | 9.4 (6.5) | 10.0 (7.5) | –* | 0.281 |
| GP | 5.7 (4.5) | 6.2 (6.0) | 7.5 (6.5) | <0.001 |
| GP out of hours service* | 0.1 (0.3) | 0.2 (0.9) | –* | 0.118 |
| Practice nurse/practitioner/HCA* | 3.2 (3.3) | 2.8 (2.9) | –* | 0.161 |
| Community nurse* | 0.5 (2.0) | 0.8 (2.0) | –* | 0.473 |
| Clerical | 0.3 (0.7) | 0.4 (1.8) | 5.8 (10.7) | <0.001 |
| Pharmacist/pharmacy technician | 0.1 (0.3) | 0.0 (0.2) | 0.0 (0.3) | 0.448 |
| All other consults | 0.2 (0.6) | 0.4 (1.1) | 0.0 (0.1) | <0.001 |
| Inpatient | 2.4 (9.9) | 3.5 (11.5) | 7.4 (18.6) | 0.010 |
| Day patient | 0.2 (0.7) | 0.2 (0.6) | 0.1 (0.4) | 0.373 |
| Outpatient‡ | 0.5 (1.0) | 0.5 (1.2) | 0.6 (1.9) | 0.069 |
| A&E‡ | 0.1 (0.3) | 0.1 (0.4) | 0.1 (0.4) | 0.896 |

*Not available at 60 months.
†P value for change over time.
‡Numbers based on 3 months prior to interview.
A&E, accident and emergency services; GP, general practitioner; HCA, healthcare assistant.

medical workforce is adequately equipped to meet the needs of our ageing population. Strangely, geriatric experience is not a core part of GP training, and clinical teaching in this area within undergraduate medical curricula is limited.[15] It is interesting to note that recent national recommendations to extend core GP training in the UK from 3 years to 4 years, with a focus on the management of age-related issues such as multimorbidity, frailty and cognitive impairment and dementia, remain as recommendations and have not been translated into practice.[16] Although GP training and primary healthcare provision vary between countries, ageing is a global issue and there are already concerns that current specialist-led models of care provision are not sustainable to meet future demand.[17] Thus, while increased geriatric training for GPs may help, other issues inherent within healthcare systems need to be addressed such as the location of specialist geriatric teams, which may be more appropriately placed within community care rather than hospital services, and how GPs are rewarded or reimbursed for providing such complex and challenging care.[18]

In a majority of high-income countries, general or family practice is the mainstay of healthcare, providing first-line contacts and acting as gatekeeper to secondary care.[19] Our findings add further weight to the growing concern that the National Health Service (NHS) primary care will struggle to meet the needs of a rapidly ageing population, in the face of declining GP recruitment.[20 21]

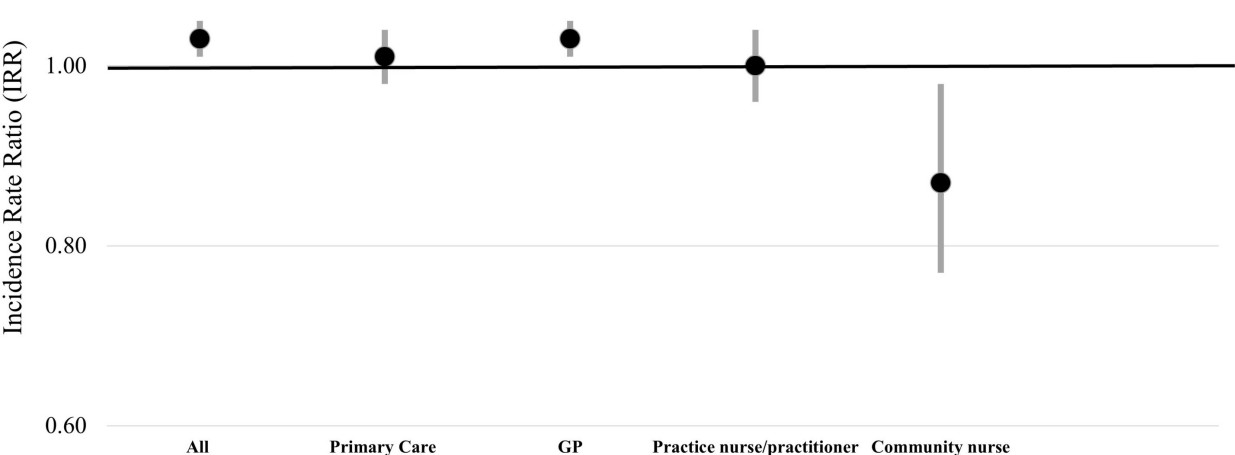

**Figure 1** Time trends in primary and community care consultations (IRR and 95% CI) adjusted for sex, living status, self-rated health, Mini-Mental State Examination, geriatric depression score and disease groups count. Primary care, practice nurse\practitioner and community nurse analysed between baseline and 36 months. GP, general practitioner; IRR, incidence rate ratio.

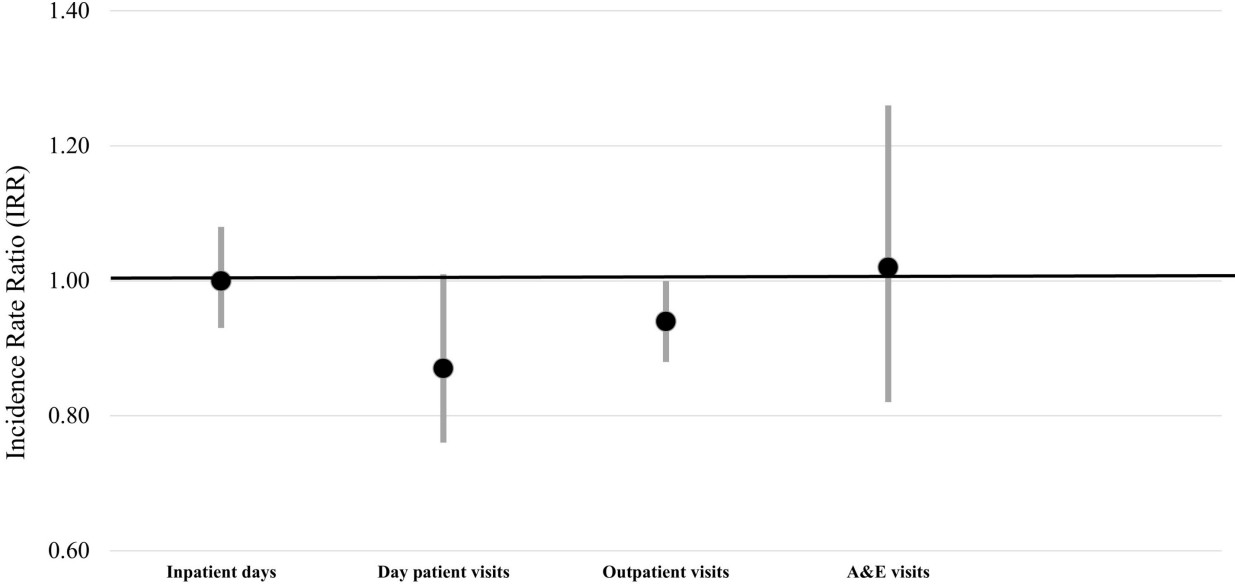

**Figure 2** Time trends in secondary care consultations (IRR and 95% CI) adjusted for sex, living status, self-rated health, Mini-Mental State Examination, geriatric depression score and disease groups count. A&E accidents and emergency; IRR, incidence rate ratio.

Recent research, looking at over 100 million primary care consultations for all age groups between 2007 and 2014, found that GP workload rose by more than 16% compared with <1% for practice nurses[14]; consultations rates were highest for the very young (<4 years) and the very old (85 years). The authors concluded that such an increase was probably an underestimate, as the data excluded other GP duties such as administration and teaching. They also found that GP consultations were becoming longer. In England, an average GP consultation is 10 min, but longer for people aged over 65 years.[22] For people aged 85 and over, where there are high rates of sensory impairment[3 5] and multimorbidity is the norm, such consultations may

be longer and more complex. The skills required may explain the importance of the GP as healthcare provider to this population, despite the rapidly increasing role of nurses and nurse practitioners in primary care.[18]

The number of nursing and residential home is decreasing,[23] while the number of older people with significant care needs living at home is increasing.[24] This combination can only increase the pressure on primary and community care services,[18 22] while continued financial austerity requires increased cost efficiency in service provision. Better access to geriatric expertise, through community-based multidisciplinary assessment teams, may in future be beneficial to both patients and our

primary gatekeeper healthcare services by providing the latter with easier access to specialist knowledge and support.[18 22] Although our findings currently reveal the GP as the key care provider for the very old, the crisis in recruitment of doctors suggest that the potential of specialist nurse practitioners to improve patient and care outcomes should be considered. Whether such a service would be acceptable to older people as an alternative to seeing the GP requires further exploration, but the integration of specialist palliative care nurses into routine NHS care provide an encouraging precedent.[25 26]

In summary, if GPs are the central care provider for our older people, they must be knowledgeable, skilled and better supported by appropriately located specialist services to ensure that our medical workforce is equipped to meet the needs and demands of a 21st century ageing population. In addition to the inclusion of geriatrics in GP training, the provision of such teaching within medical undergraduate curricula needs to be urgently reviewed, in terms of the nature, content and timing of such teaching, in order that future generations of doctors, not just GPs, are adequately prepared. Finally, future research is required to explore how best to configure services to address the healthcare needs of older people while maintaining quality of care; such studies must include the very old, a subgroup often neglected from research trials, to ensure their future care is truly evidence based.[27]

**Acknowledgements** We would like to thank the 85-year-olds of Newcastle and North Tyneside, and their families and carers, for the generous donation of their time and personal information. In addition, we thank the research nurses, data manager, project secretary and the North of England Commissioning Support Unit and local general practitioners and their staff.

**Contributors** LR conceived the study, obtained project funding and drafting of the paper. MEY, CJ and TF were responsible for data analysis and drafting of the paper. RD, BH and SP contributed to drafting of the paper. All authors approved the final manuscript.

**Funding** This work was supported by UK MRC and BBSRC (G0500997), Dunhill Medical Trust (R124/0509); Newcastle Healthcare Charity; NIHR Newcastle Biomedical Research Centre. This paper presents independent research funded by the National Institute for Health Research School for Primary Care Research (NIHR SPCR), project number SPCR 303. LR is funded by a National Institute for Health Research Professorship (NIHR-RP-011-043).

**Disclaimer** The views expressed are those of the author(s) and not necessarily those of the NIHR, the National Health Service or the Department of Health.

**Competing interests** None declared.

**Patient consent** Obtained.

**Ethics approval** Newcastle and North Tyneside 1 Research Ethics Committee (reference number 06/Q0905/2)

**Provenance and peer review** Not commissioned; externally peer reviewed.

**Data sharing statement** Newcastle 85+ study data may be obtained by agreement from the Data Guardians Group on submission of a data request form (available at: https://research.ncl.ac.uk/85plus/datarequests).

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
