## [Reviewer comments · BMJ Open]

ARTICLE DETAILS

TITLE (PROVISIONAL)	Use of primary care and other healthcare services between age 85 and 90: longitudinal analysis of a single year birth cohort, the Newcastle 85+ Study.
AUTHORS	Yadegarfar, Mohammad; Jagger, Carol; Duncan, Rachel; Fouweather, Tony; Hanratty, Barbara; Parker, Stuart; Robinson, Louise

VERSION 1 – REVIEW

REVIEWER	Zhe He Florida State University, USA
REVIEW RETURNED	07-Sep-2017

GENERAL COMMENTS	This is a short report about the use of primary care services by very old adults in a cohort study. The motivation of this study is clear. The statistical methods were properly used in the data analysis. The results were concisely described and explained. The discussion section is informative. The reviewer only found a few issues with this paper: 1. It is intuitive that the financial status of older adults is associated with their healthcare. It will be nice if finance status can be added as an independent variable.2. Family support should also be considered as an independent variable in the data analysis.3. Disease group should be elaborated. Which diseases were included?4. In the abstract, the full form of the acronym should be given in the first appearance.
--

REVIEWER	Rosaly Correa-de-Araujo National Institute on Aging, National Institutes of Health, USA
REVIEW RETURNED	08-Sep-2017

GENERAL COMMENTS	This is an interesting study which focuses on the use of primary care and other healthcare services by older adults > 85 years of age. The strength of this study relies on the analyses of general practitioners' medical records containing information on the use of such services by this population group in the Newcastle cohort study. The manuscript is well written but it requires some revision to define abbreviations that may have different meaning in older parts of the globe. Examples include A&E, HCA, NHS. These should be spelled out at least at their first appearance in the text. The methodological approach is scientifically sound. The findings related to the very old being more likely to use general practitioners compared to other primary healthcare professionals is indeed a matter of concern not only due to limited knowledge and experience of general practitioners on how to care for older adults. This reviewer recommends that authors:  • Emphasize how critical training is in view of current limitations with inclusion of older adults, and in particular the very old, in clinical studies. This jeopardizes the provision of evidence-based care, making clinical decisionmaking challenging in a population with complex health problems such as multiple chronic conditions. • Emphasize that geriatrics training should start early in medical school curriculum. • Expand their discussion to highlight other barriers (e.g., care coordination with a multidisciplinary team, financial resources) that also affect general practitioners' ability to provide better care. • Explain that while geriatric training may help overcome these barriers, policy changes are needed to better define the role of general practitioners in the context of the health care system including issues related to their reimbursement. • Clarify that general practitioners may use different approaches (depending on country and culture). Some focus on the health of the whole person – physical, psychological, and social, which is a plus in the care of older adults. Therefore, expanding and enhancing geriatric care training for these professionals could help integrate general practitioners in multidisciplinary health care teams and considerably improve health outcomes for older adults. The manuscript could also be enriched by displaying reliable geriatrics training resources for general practitioners. This reviewer recommends that revisions be made accordingly prior to publication to ensure a more attractive and useful study. As for future research, authors may consider analyzing the Newcastle cohort to look into impact of general practitioners' care on health outcomes for the very old.
--

VERSION 1 – AUTHOR RESPONSE

Reviewer #1:

This is a short report about the use of primary care services by very old adults in a cohort study. The motivation of this study is clear. The statistical methods were properly used in the data analysis. The results were concisely described and explained. The discussion section is informative. The reviewer only found a few issues with this paper:

1. It is intuitive that the financial status of older adults is associated with their healthcare. It will be nice if finance status can be added as an independent variable.

Author's response:

We thank the reviewer for these comments. In the UK healthcare is free at the point of contact and therefore is not generally related to financial status. We did include education in the models as a confounding factor and in this cohort education is closely related to deprivation.

2. Family support should also be considered as an independent variable in the data analysis.

Author's response:

We have already included living status (living alone, with others, in institutional care) as a confounding factor. Family support will be highly correlated with this.

3. Disease group should be elaborated. Which diseases were included?

Author's response:

The diseases included in the disease grouping are specified in Table 1.

Action:

We have added a footnote in Table 2 to signpost to Table 1.

4. In the abstract, the full form of the acronym should be given in the first appearance.

Action:

We have been through the manuscript and ensured that the full form of any acronyms are given at first appearance.

Reviewer #2:

1. The manuscript is well written but it requires some revision to define abbreviations that may have different meaning in older parts of the globe. Examples include A&E, HCA, NHS. These should be spelled out at least at their first appearance in the text.

Author's response:

We thank the reviewer for these comments and reiterate that we have checked that all acronyms are given in full at first appearance.

2. The findings related to the very old being more likely to use general practitioners compared to other primary healthcare professionals is indeed a matter of concern not only due to limited knowledge and experience of general practitioners on how to care for older adults.

This reviewer recommends that authors:

- Emphasize how critical training is in view of current limitations and emphasize that geriatrics training should start early in medical school curriculum.

Author's response:

The discussion section has been re-written to include this with a second paragraph focused on the importance and urgency of issues around both GP and undergraduate medical training both in the UK and internationally (page 8).

- Explain that while geriatric training may help overcome these barriers, policy changes are needed to better define the role of general practitioners in the context of the health care system including issues related to their reimbursement.
- Expand their discussion to highlight other barriers (e.g., care coordination with a multidisciplinary team, financial resources) that also affect general practitioners' ability to provide better care.

Author's response:

As outlined above, the revised discussion section include a new paragraph on both medical training and also consideration of other barriers that currently influence the provision of high quality care to older people such as access to specialist services to support GPs and current financial reimbursement (pages 8-9).

- Clarify that general practitioners may use different approaches (depending on country and culture). Therefore, expanding and enhancing geriatric care training for these professionals could help integrate general practitioners in multidisciplinary health care teams and considerably improve health outcomes for older adults. The manuscript could also be enriched by displaying reliable geriatrics training resources for general practitioners.

Author's response:

Whilst we agree with the reviewer on this point, we felt it difficult to address as there are limited globally appropriate training resources for GPs with most being language or country specific.

- Emphasize how critical training is in view of current limitations with inclusion of older adults, and in particular the very old, in clinical studies. This jeopardizes the provision of evidence-based care, making clinical decision making challenging in a population with complex health problems such as multiple chronic conditions.

Author's response:

Whilst addressing the above comments on the importance of changes to training, policy and practice, we have also now acknowledged reviewer 2's point about the need to include the very old in research studies, especially RCTs of new service interventions, to ensure their future care is appropriate evidenced-based (page 10).

VERSION 2 – REVIEW

REVIEWER	Zhe He Florida State University
REVIEW RETURNED	11-Nov-2017

GENERAL COMMENTS	The authors have adequately addressed the comments. I have no further comments to make.
---

REVIEWER	Rosaly Correa-de-Araujo National Institutes of Health, National Institute on Aging, United States
REVIEW RETURNED	11-Nov-2017

GENERAL COMMENTS	This reviewer feels the authors properly addressed the reviewer's comments and the paper reads much better and is more informative. This reviewer recommends that the authors continue to monitor access to and utilization of services by older adults, as well as their inclusion of this population in clinical trials or other types of studies. One area of concern relates to quality of health services received and this is linked to the availability of the evidence. Future papers addressing these issue are needed.
--